# Cholesteryl Hemiazelate Present in Cardiovascular Disease Patients Causes Lysosome Dysfunction in Murine Fibroblasts

**DOI:** 10.3390/cells12242826

**Published:** 2023-12-13

**Authors:** Elizeth Lopes, Gisela Machado-Oliveira, Catarina Guerreiro Simões, Inês S. Ferreira, Cristiano Ramos, José Ramalho, Maria I. L. Soares, Teresa M. V. D. Pinho e Melo, Rosa Puertollano, André R. A. Marques, Otília V. Vieira

**Affiliations:** 1iNOVA4Health, NOVA Medical School, Faculdade de Ciências Médicas, Universidade Nova de Lisboa, 1150-069 Lisbon, Portugal; elizeth.lopes@nms.unl.pt (E.L.); giselasofia@hotmail.com (G.M.-O.); catarina.simoes@nms.unl.pt (C.G.S.); ines.s.ferreira@nms.unl.pt (I.S.F.); ramos.cristiano.93@gmail.com (C.R.); jose.ramalho@nms.unl.pt (J.R.); 2Coimbra Chemistry Centre (CQC)–Institute of Molecular Sciences and Department of Chemistry, University of Coimbra, 3004-535 Coimbra, Portugal; misoares@ci.uc.pt (M.I.L.S.); tmelo@ci.uc.pt (T.M.V.D.P.e.M.); 3Cell and Developmental Biology Center, National Heart, Lung, and Blood Institute, National Institutes of Health (NIH), Bethesda, MD 20892, USA; puertolr@nhlbi.nih.gov

**Keywords:** mouse embryonic fibroblasts, atherosclerosis, cholesteryl hemiesters, cholesteryl hemiazelate, lysosome dysfunction, autophagy and apoptosis

## Abstract

There is growing evidence supporting the role of fibroblasts in all stages of atherosclerosis, from the initial phase to fibrous cap and plaque formation. In the arterial wall, as with macrophages and vascular smooth muscle cells, fibroblasts are exposed to a myriad of LDL lipids, including the lipid species formed during the oxidation of their polyunsaturated fatty acids of cholesteryl esters (PUFA-CEs). Recently, our group identified the final oxidation products of the PUFA-CEs, cholesteryl hemiesters (ChE), in tissues from cardiovascular disease patients. Cholesteryl hemiazelate (ChA), the most prevalent lipid of this family, is sufficient to impact lysosome function in macrophages and vascular smooth muscle cells, with consequences for their homeostasis. Here, we show that the lysosomal compartment of ChA-treated fibroblasts also becomes dysfunctional. Indeed, fibroblasts exposed to ChA exhibited a perinuclear accumulation of enlarged lysosomes full of neutral lipids. However, this outcome did not trigger de novo lysosome biogenesis, and only the lysosomal transcription factor E3 (TFE3) was slightly transcriptionally upregulated. As a consequence, autophagy was inhibited, probably via mTORC1 activation, culminating in fibroblasts’ apoptosis. Our findings suggest that the impairment of lysosome function and autophagy and the induction of apoptosis in fibroblasts may represent an additional mechanism by which ChA can contribute to the progression of atherosclerosis.

## 1. Introduction

Atherosclerosis and its clinical manifestations, myocardial infarction, and ischemic stroke are the leading causes of death and disability worldwide [1]. This pathology is characterized by lipid accumulation in the subendothelial space, intimal inflammation, vascular smooth muscle cells (VSMCs) migration from the media to the outside of the newly formed plaque and ultimately plaque rupture [2]. Besides endothelial cells, macrophages and VSMCs, an additional cell type, the fibroblast, is also an important player in atherosclerosis.

Fibroblasts are abundant and very plastic cells in the vasculature. Single cell sequencing analysis has shown that although VSMCs comprise the largest cell population in the murine aorta (40%), fibroblasts constitute roughly 33% of the aortic cells [3]. Furthermore, single cell sequencing analysis has revealed different fibroblast subsets in healthy and atherosclerotic tissue [3] (for recent reviews, see [4,5]). Fibroblasts seem to be involved in all stages of atherosclerosis, from the initial phase to fibrous cap and plaque formation. Accordingly, adventitial fibroblasts are some of the first cells to respond and become activated in the initial stage of atherosclerosis, even before the formation of atherosclerotic lesions [5,6,7]. Indeed, during the initial stages of atherosclerosis, a substantial number of adventitial fibroblasts can differentiate into activated fibroblasts (myofibroblasts) in response to proinflammatory signals and extracellular matrix (ECM) remodeling, among other cues [8,9]. Myofibroblasts intensely proliferate and, as in all chronic inflammatory conditions, synthesize and release large amounts of ECM proteins, growth factors, proinflammatory cytokines, chemokines and proteolytic enzymes. Extensive remodeling through the activity of matrix metalloproteinases and an increase in their inhibitors, as well as reactive oxygen species, result in tissue damage [8,10,11,12].

Furthermore, these secreted molecules directly affect the phenotypes of other resident cells in the vessel wall, such as VSMCs and endothelial cells, promoting neointima formation, regulating vasa vasorum expansion and affecting the recruitment of infiltrating immune cells [13,14,15,16]. Thus, the role of fibroblasts in atherosclerosis is complex and multifaceted, and their involvement is not limited to being solely proatherogenic. Fibroblasts, by being recruited to the plaque site and by producing collagen, which strengthens the fibrous cap over the atherosclerotic plaques, represent a protective process, preventing plaque rupture [10]. Accordingly, it was recently discovered that the inhibition of the expression or activity of fibroblast activation protein (FAP), which is expressed in activated but not in quiescent fibroblasts and found to be associated with atherosclerotic plaques, can attenuate the progression of atherosclerosis by increasing plaque stability in experimental mouse models of atherosclerosis [17].

Atherosclerosis has long been identified as an inflammatory disease with low-density lipoprotein (LDL) overload. Adequate lipid catabolism has the effect of delaying the progression of atherosclerosis. However, excessive lipid accumulation, mainly due to the formation of undigestible oxidized LDL lipid species, in lysosomes may cause the progressive dysregulation of lysosomes and autophagy function. Therefore, lysosomes present the characteristics of a larger volume and decreased protein and lipid degradation ability, which aggravates the deterioration of the plaque microenvironment and accelerates plaque rupture [18,19]. Recently, our lab discovered a novel family of oxidized lipids in the plasma of cardiovascular disease patients and in carotid atheroma plaques [20,21]. These lipids, cholesteryl hemiesters (ChE), are the oxidation products of the LDL polyunsaturated fatty acids of cholesteryl esters. Importantly, the most prevalent ChE detected in human plasma, cholesteryl hemiazelate (ChA), is sufficient to cause lysosome dysfunction in murine macrophages and VSMCs [22,23], and in macrophages in zebrafish larvae [21]. We have already established that ChA accumulates in lysosomes and impairs lysosome acidification, hydrolase activity and cargo egress, impacting the lysosomal membrane composition and intracellular localization [22,23]. Furthermore, in zebrafish larvae fed with a ChA-enriched diet, the lysosome membrane integrity was compromised [21]. Lysosome malfunction has cellular consequences. In our experimental models, lysosome malfunction culminated in pathological foam cell formation, inflammation and an increase in VSMC stiffness [21,22,23]. In this context, we decided to assess whether ChA can induce lysosomal dysfunction and the loss of cell homeostasis in a murine model of fibroblasts that, like macrophages and VSMCs, are exposed to this lipid in the arterial wall during atheroma formation.

Here, we show that ChA is sufficient to induce lysosomal dysfunction in fibroblasts. However, the adaptation of these cells to lysosome stress does not trigger lysosome biogenesis. As a consequence, autophagy is inhibited, and the apoptosis of the fibroblasts is observed. These results have been already described to occur in macrophages and VSMCs exposed to oxidized LDL (oxLDL) [24,25,26,27].

## 2. Materials and Methods

### 2.1. Liposome Preparation

In this study, 1-palmitoyl-2-oleoyl-glycero-3-phosphocholine (POPC, Avanti Polar Lipids, Alabaster, AL, USA) was used as a vehicle, and the details for the preparation of POPC liposomes were previously described in [22]. Cholesteryl hemiazelate (ChA) was synthesized at the Department of Chemistry, University of Coimbra, Coimbra, Portugal. We prepared ChA:POPC liposomes with a molar ratio of 65:35, as previously described [22]. To determine its concentration, we performed the Liebermann protocol [28].

### 2.2. Cell Culture and Lipid Treatment

Mouse embryonic fibroblasts (MEFs) wild type (WT; control cell line) and TFEB/3-DKO cells were generated by CRISPR/Cas9 technology, as described in [29]. Cells were grown in Dulbecco’s modified Eagle medium (DMEM Glutamax, Gibco, Thermo Fisher, Waltman, MA, USA), supplemented with 10% heat-inactivated fetal bovine serum (FBS, Gibco), 1% sodium pyruvate (Gibco) and 1% PenStrep (10,000 units/mL penicillin, 10,000 μg/mL streptomycin (Gibco)). Cells were maintained at 37 °C with 5% CO_2_ in a humidified incubator, and they were passaged twice a week using 1X Trypsin–Ethylenediaminetetraacetic acid (EDTA, 0.05%, Gibco) for detachment. For experiments, cells were seeded for 24 h and subsequently treated with 269.2 μM of POPC (equivalent POPC concentration present in 500 μM of ChA) and 500 μM of ChA, for different time points, as indicated in the figure legends.

### 2.3. Fluorescence Microscopy, Staining, Image Acquisition and Analysis

MEFs WT and TFEB/3-double knock-out (DKO) cells were seeded, and, after 24 h, cells were incubated with POPC and ChA for 72 h. Cells were washed with phosphate-buffered saline (PBS) 1X and fixed in 4% paraformaldehyde (PFA) for 20 min. Ammonium chloride 10 mM was used to quench the aldehyde groups and 0.1% saponin in PBS 1X was used to permeabilize the cells, followed by blocking with 2% gelatine or 1% BSA in PBS. Cells were incubated overnight with the following primary antibodies: rat anti-LAMP1 (Hybridoma Bank, 1D4B), rabbit anti-TFEB (Bethyl Laboratories, A303-673A) and rabbit anti-TFE3 (Sigma, Atlas Antibodies, St. Louis, MO, USA, HPA023881). This was followed by 1 h of incubation with the secondary antibodies conjugated with Cy3 and Cy5 (Jackson ImmunoResearch Laboratories, Baltimore Pike, PA, USA). Bodipy 493/503 (Invitrogen, Carlsbad, CA, USA, D3922) and DAPI (Fluka, Buchs, Switzerland, 2491867) were used to label neutral lipids and the nuclei, respectively, for 1 h. Finally, coverslips were mounted with Dako Fluorescence Mounting Medium (Dako, Santa Clara, CA, USA), and images were obtained using an AxioVision microscope (Axio Observer Z2, Jena, Germany) with a 63x NA-1.4 oil immersion objective.

### 2.4. Western Blotting

Cell lysates were prepared using a mix of RIPA lysis buffer [23], protease (cOmplete, EDTA-free, Mannheim, Germany, 1187580001) and phosphatase (Calbiochem, Darmstadt, Germany, 524625) inhibitors. The protein concentration was determined using the PierceTM BCA Protein Assay Kit (Thermo Scientific, Rockford, IL, USA, 23225). Samples were then mixed with sample buffer (5X Laemmli Buffer: Tris–HCl 250 mM pH 6.8, 10% SDS, 50% glycerol, 25% 2-mercaptoethanol, 0.005% bromophenol), heated for 5 min at 95 °C, and 20–50 µg were loaded into a 10 or 15% SDS polyacrylamide gel. After electrophoresis, proteins were transferred toward an activated PVDF membrane in 1x transfer buffer at 4 °C for 1 h. Membranes were blocked with blocking buffer (5% BSA or milk in 1X TBS-T–Tris-buffered saline, 0.01% tween) for 1 h at room temperature (RT), followed by overnight incubation at 4 °C with the following primary antibodies diluted in blocking buffer: rat anti-LAMP1 (Hybridoma Bank, 1D4B), rabbit anti-CTSD (Abcam, ab75852), rabbit anti-LAL (Abcam, Cambridge, UK, ab154356), mouse anti-P62 (Abgent, San Diego, CA, USA, AP2183b), rabbit anti-LC3B (Cell Signaling, Danvers, MA, USA, 2775S), rabbit anti-phospho-mTOR (Cell Signaling, 5536S), rabbit anti-mTOR (Cell Signaling, 2983S), 4E-BP1 (Cell Signaling, 9644S), phosphor-4E-BP1 (Cell Signaling, 2855S), rabbit anti-S6 ribosomal protein (Cell Signaling, 2217S), rabbit anti-phospho-S6 (Cell Signaling, 4858S), rabbit anti-caspase 3 (Cell Signaling, 9661S), rabbit anti-p16-INK4A (Proteintech, Rosemont, IL, USA, 10883-1-AP), mouse anti-p21 (Santa Cruz, Dallas, Texas, United States, sc-6246), mouse anti-SQSTM1 (Abgent, AP2183b), mouse anti-CD36 (BD Biosciences, San Jose, CA, USA, 552544), rabbit anti-SR-BI (NOVUS Biologicals, Littleton, CO, USA, NB400-113), rabbit anti-CD204 (Abcam, ab151707), TLR4 (Invitrogen, Carlsbad, CA, USA, 48-2300), goat anti-GAPDH (SICGEN, AB0049-20), goat anti-calnexin (SICGEN, Cantanhede, Portugal, AB0041-20), mouse anti-β-tubulin (Hybridoma Bank, Iowa City, IA, USA, E7). After incubation with primary antibodies, membranes were washed with 1X TBS-T and incubated for 1 h at RT with the corresponding horseradish peroxidase secondary antibody diluted in blocking buffer. Blots were visualized using the Clarity^TM^ western ECL substrate (Bio-Rad laboratories, Hercules, CA, USA, 170-5061) in a Chemidoc Touch Imaging System (Bio-Rad Laboratories). Bands were quantified using the Fiji software, v1.53f51.

### 2.5. LDH Assay

MEFs WT or TFEB/3-DKO cells were seeded, and, 24 h later, they were treated either with ChA or POPC liposomes for 96 h. To assess LDH activity, we performed the LDH activity assay, according to the manufacturer’s instructions, using the CyQUANT™ LDH Cytotoxicity Assay Kit (Invitrogen, Eugene, OR, USA, C20300). Samples with the maximum LDH activity were diluted four times to ensure that the absorbance readings fell within the linear range of the assay. The absorbance was measured in a Synergy HT plate reader at 490 nm and 680 nm, and LDH activity was determined by subtracting the 680 nm absorbance value (background) from the 490 nm absorbance before the calculation of the percentage (%) of LDH activity in the cell culture media relative to the control (WT POPC).

### 2.6. Total and Free Cholesterol and Cholesteryl Ester Measurement

Total and free cholesterol and cholesteryl ester levels in cell lysates were determined using an Amplex Red-based Commercial Kit (A12216, Amplex™ Red Cholesterol Assay Kit, ThermoFisher). MEFs WT or TFEB/3-DKO were seeded and, 24 h later, they were treated with either ChA or POPC liposomes for 96 h. Cells were washed three times in PBS and lysed in water [23]. Cholesterol levels were normalized to the cell protein levels, determined using the PierceTM BCA Protein Assay Kit. Fluorescence was measured in a Synergy HT plate reader with excitation at 545 nm and emission at 590 nm.

### 2.7. qRT-PCR

MEFs WT and TFEB/3-DKO were treated with lipids (POPC and ChA) for 24 h, 48 h, 72 h and 96 h and the total RNA was extracted with the NZY Total RNA Isolation Kit (Nzytech, Lisbon, Portugal, MB13402); then, reverse transcription was performed using the NZY First-Strand cDNA Synthesis Kit (Nzytech, MB12501) according to the manufacturer’s instructions. Quantitative PCR was performed in a 96-well plate using NZYSupreme qPCR Green Master Mix (Nzytech, MB44003) in a QuantStudio™ 5 Real-Time PCR System (Thermo Fisher). GAPDH and PGK1 were used as housekeeping genes to normalize gene expression. Target gene expression was determined by relative quantification (2^−ΔΔCt^ method) to the housekeeping reference gene. The primers were ordered from Thermo Fisher and the sequences are indicated in Table 1.

### 2.8. LAL Activity Assay

The LAL activity was assessed as previously described in [22]. Briefly, MEFs WT and TFEB/3-DKO were seeded in a 6-well plate and, after 24 h, they were incubated with POPC and ChA for 72 h. Cells were washed twice with PBS and lysed in a mix of LAL reaction buffer and protease inhibitors for 30 min at 4 °C, rotating, and cleared by centrifugation for 10 min at 16,000× *g*. The protein concentration of the supernatant was determined using the PierceTM BCA Protein Assay Kit. Substrate mix Lalistat2 (Sigma, SML2053) was added, followed by incubation for 90 min at 37 °C. Then, a stop buffer (150 mM EDTA pH 11) was added, and fluorescence was measured in a plate reader, the Tecan Infinite F200 PRO, with excitation at 360 nm and emission at 465 nm.

### 2.9. Endocytosis Evaluation by Flow Cytometry

MEFs WT and TFEB/3-DKO cells were seeded in a 24-well plate for 24 h. The medium was then removed, the cells were washed once with PBS 1X and 300 µL of fresh medium without serum was added, followed by the addition of 200 µg/mL of Texas Red™-conjugated Bovine Serum Albumin (BSA, Invitrogen, A23017). To prevent BSA degradation, 20 mM NH_4_Cl was added 20 min after incubation to neutralize the lysosome pH. After incubation, the cells were washed three times with PBS 1X, detached using FACS buffer or PBS-BSA (1%) and centrifuged twice at 300× *g* for 5 min at RT. The supernatant was discarded, and the cells were washed twice with PBS 1x. The cell pellet was then resuspended in FACS buffer or PBS-BSA (1%) and transferred to FACS tubes for analysis. The samples were read using FACS Aria III-PE-mCherry.

### 2.10. Micropinocytosis Evaluation by Flow Cytometry

MEFs WT and TFEB/3-DKO were seeded and, after 24 h, they were incubated with POPC, ChA and 200 µg/mL of dextran–fluorescein (Invitrogen, D1820) for another 24 h. Then, the cells were washed with PBS 1X and detached by incubation with trypsin for 5 min at 37 °C. Cell suspensions were centrifuged at 300× *g* for 5 min at RT, the supernatants were discarded, the cells were fixed with 4% PFA on ice for 15 min and then centrifuged at 300× *g* for 5 min at RT. The cells were washed with PBS 1X and centrifuged at 300× *g* for 5 min at RT; the cell pellet was resuspended in FACS buffer and then transferred to FACS tubes for analysis. The samples were read using FACS Canto II.

### 2.11. Statistical Analysis

Data are representative of at least three independent experiments and values on the graphs are indicated as the mean ± SEM. Statistical significance was assessed by the GraphPad Prism software v. 9.0.0 using a *t*-test or one-way ANOVA–Kruskal–Wallis test depending on the data distribution. *p* < 0.05 (*), *p* < 0.01 (**), *p* < 0.001 (***) and *p* < 0.0001 (****) were considered statistically significant.

## 3. Results

### 3.1. Cholesteryl Hemiazelate Is Sufficient to Cause Lysosome Dysfunction in Mouse Embryonic Fibroblasts

Fibroblasts are abundant cells in aortic vessels, both in humans and in animal models [4]. During atheroma formation, fibroblasts, like VSMCs and macrophages, are exposed to miscellaneous molecules, including lipids formed during LDL oxidation. Our previous publications have shown that ChA, an end-product of the oxidation of the cholesteryl esters of LDL particles and the most prevalent ChE in the tissues of cardiovascular disease patients [20,21], negatively impacts lysosome function in both VSMCs and macrophages, with consequences for their homeostasis [22,23]. Thus, in this study, we decided to assess the ability of ChA to alter lysosome function in fibroblasts and consequently the possible stress responses induced on these cells. As a model of fibroblasts, we used mouse embryonic fibroblasts (MEFs). As before [22,23], all experiments were performed with ChA:phosphatidylcholine (POPC) liposomes (65:35, molar ratio). Liposomes were used as ChA vehicles to avoid artefacts due to the possible uptake of microcrystals of ChA, which can be formed due to the poor solubility of this lipid in the cell culture medium. POPC liposomes were used as a vehicle control.

The impact of ChA on lysosomes in MEF wild-type (WT) cells was initially assessed by monitoring the morphology and lysosome number (Figure 1A,C,D). For this purpose, MEFs were treated with ChA and lysosomes were labelled by staining with antibodies targeting the lysosome-associated membrane protein-1 (LAMP1). LAMP1 is a marker of late endosomes and lysosomes (LE/Lysosomes, hereafter collectively referred to as lysosomes, unless stated otherwise). As shown in Figure 1A,C (see insets), ChA causes a statistically significant enlargement in the area of lysosomes near the perinuclear region, when compared with the lysosome area in POPC-treated cells. The mean lysosomal area in ChA-treated MEFs was 1.099 ± 0.4216 µm^2^ in comparison to 0.7832 ± 0.1696 µm^2^ in control cells (Figure 1C, *n* = 20). Importantly, the microscopy images also showed that MEFs incubated with ChA exhibited some neutral lipid accumulation, observed by the increase in the signal of the fluorescent neutral lipid Bodipy 493/503 dye, inside LAMP1-immunostained enlarged lysosomes. This outcome is probably a consequence of lysosome dysfunction, culminating in the accumulation of undegraded endocytosed neutral lipids that exist in the cell culture medium [22,23]. In contrast, in POPC-exposed cells, there were no detectable BODIPY-positive lysosomes (LAMP1-positive structures) and only a limited number of small lipid droplets (LAMP1-negative structures, Figure 1A, upper panels), indicating that neutral lipids are not being stored under these conditions. However, lysosome enlargement and neutral lipid accumulation in their lumen were not followed by an increase in lysosome number (Figure 1D), nor by an increase in lysosome mass, which was assessed by the total LAMP1 protein levels (Figure 1E,F) [30].

Under lysosomal stress, the cells adapt by activating the transcription factor EB (TFEB) and the related transcription factor E3 (TFE3), which in turn engage a gene network containing >400 genes, many of which encode lysosomal proteins [29,31]. Therefore, we decided to assess whether TFEB and TFE3 were involved in lysosome enlargement induced by ChA. For this purpose, the set of experiments described above was also performed in TFEB and TFE3 double knockout (TFEB/3 DKO) MEFs. In these cells, ChA did not induce any perceptible changes at the level of the lysosome mean area (Figure 1B,C), suggesting that TFEB/3 could be involved in lysosome enlargement in WT MEFs challenged with ChA. As in WT cells, the number and mass of lysosomes (assessed by LAMP-1 protein levels) were not altered by ChA in DKO MEFs (Figure 1E,F). However, we did detect a minor (slightly significant) increase in lysosome number in ChA-treated DKO cells compared to ChA-treated WT cells (Figure 1D), not reflected by the total LAMP1 protein level (Figure 1E).

Morphological alterations in lysosomes are normally associated with the impairment of their degradative capacity [32]. Furthermore, the accumulation of neutral lipids within these organelles is an indication that this could be the case. Therefore, to investigate whether the catalytic activity of the lysosomes in ChA-treated MEFs was compromised, we performed additional experiments. We started by quantifying the mature (active) and the pro-cathepsin D levels and the ratio of mature/total cathepsin D (Figure 1G–I). In our opinion, this readout is the most sensitive and robust assay to assess lysosome dysfunction. As observed in Figure 1G and quantified in Figure 1H, the total cathepsin D levels were lower in ChA-treated cells than in control cells, and roughly 50% less of this lysosomal enzyme was efficiently processed to the active (mature) form (Figure 1I). We also assessed the levels and the activity of the lysosomal acidic lipase (LAL, encoded by the *Lipa* gene) (Figure 1G,J,K). The protein levels of this enzyme were similar in ChA-treated and control cells (Figure 1J). However, its catalytic activity was reduced by ChA in a significant manner when compared to POPC-treated cells (Figure 1K). Cathepsin D and LAL are synthesized as zymogens and require proteolytic processing at a low pH to become active [33,34]. Thus, our data suggest that pro-CTSD is not being properly processed in the lysosomes to the mature (active) form of the enzyme and that ChA negatively impacts LAL activity, probably due to changes in the lysosome luminal pH [22,23]. These results indicate that the lysosomes in ChA-treated MEFs were dysfunctional, resulting in the accumulation of undigested cargo in the lumen and their subsequent enlargement. As above, ChA in MEFs without TFEB/3 did not cause any impact in terms of lysosome function (Figure 1G–K). Therefore, we next questioned whether these lysosomal transcription factors were activated by ChA.

As previously mentioned, under conditions of lysosomal dysfunction, lysosomal transcription factors (TFs) are quickly de-phosphorylated in the cytosol and translocated to the nucleus, where they activate the transcription of the CLEAR gene network, leading to the upregulation of genes involved in lysosome functioning and autophagy [29,31,35]. Furthermore, in DKO MEFs, ChA failed to induce lysosome enlargement. Taking this information into account, we studied the translocation into the nucleus of TFEB and TFE3 in WT MEFs treated with ChA or POPC for different time periods (Figure 2A–C). Torin-1, an inhibitor of mTOR activity, was used as a positive control. The nuclear translocation was assessed by immunofluorescence (IF). As shown in Figure 2A and quantified in (Figure 2B,C), no nuclear translocation was observed for TFEB or TFE3.

To confirm that ChA was affecting nuclear translocation and not the expression of these TFs, we decided to quantify the mRNA levels of *Tfeb/3* and of other lysosomal genes that are under their transcriptional regulation. The mRNA quantitation was performed by quantitative real-time RT-PCR (qRT-PCR) (Figure 2D). At 24 h incubation time, significant mRNA increases in the transcript levels of *Tfe3* and the subunit of the v-ATPase, *Atp6v0d2*, were observed. However, for the later time point, at 72 h incubation time, the expression levels decreased relative to the control cells. For the other genes analyzed (*Tfeb*, *Lamp-1*, *Ctsd*, *Ctsb*, *Mcln1* and *Lipa*), no major changes in their mRNA levels were observed between ChA-treated and control MEFs throughout the experiment. Thus, the increase in *Tfe3* mRNA levels observed at 24 h incubation time, which has been described to occur under lysosomal stress [36,37], was not correlated with its nuclear translocation (Figure 2A,C). Therefore, to ensure that this was the case, and because there was an increase in the transcription of the lysosomal gene *Atp6v0d2*, we decided to pursue the experiments in WT and TFEB/3 DKO MEFs.

### 3.2. Cholesteryl Hemiazelate Inhibits Autophagy in Mouse Embryonic Fibroblasts

Lysosomes are the endpoint of the autophagic pathway, a critical process in preserving cellular homeostasis. Furthermore, cell autophagy inhibition affects plaque stability [27,38,39,40] and autophagy genes are transcriptionally controlled by TFEB/3 [41]. Thus, next, we assessed the expression of the *Sqstm1* and *Map1lc3b* genes, encoding for proteins of the autophagy machinery, in ChA-treated cells over time by qRT-PCR. As observed in Figure 3A, ChA did not have any impact on the mRNA levels of these two autophagic genes. In the same way, the levels of p62, an autophagy receptor encoded by the *Sqstm1* gene that targets proteins for selective autophagy, did not change with the ChA treatment relative to the control cells (Figure 3B,C). However, when the impact of ChA treatment on autophagy was evaluated, at 96 h incubation time, by quantifying the levels of the microtubule-associated proteins 1A/1B light chain 3B (LC3B, encoded by the *Map1lc3b* gene; hereafter denoted LC3) in particular, the levels of membrane-associated (lipidated) LC3 (denoted LC3-II), commonly used as a marker for autophagosome biogenesis [42], were decreased relative to the cytosolic LC3 levels (denoted LC3-I) (Figure 2B,D). This indicates that there was a suppression of the conversion of LC3-I to LC3-II, i.e., autophagy inhibition, in ChA-treated MEFs. The inhibition of autophagy upon ChA treatment was lost in TFEB/3 DKO cells, indicating that these TFs play a role in autophagy induction as well.

One of the pathways that is involved in autophagy and in TFEB/3 activation is the mammalian target of rapamycin complex 1 (mTORC1) pathway. When mTORC1 is in the active state, TFEB/3 are kept phosphorylated (inactive form) in the cytoplasm [43,44]. Likewise, the Unc-51-like autophagy activating kinase 1 (ULK1) complex, which plays a central role in the initiation stage of autophagy, is also kept in the inactive state [45]. In this context, we decided to evaluate the activation status of mTOR by Western blot (Figure 3E–H). As a control, we used the mTOR inhibitor Torin-1. As visualized in Figure 3E and quantified in Figure 3F, the ratio of p-mTOR/mTOR did not change with the ChA incubation and was similar to that in the control cells. However, the mTOR substrate 4E-BP1 was significantly more phosphorylated (activated) in ChA-treated MEFs than in the control cells (Figure 3E,G). This trend was not observed for S6K, another mTOR effector (Figure 3E,H). The fact that 4E-BP1, an mTORC1 downstream substrate and mTOR pathway activation marker, was activated could explain, at least in part, the weak activation of the lysosomal transcription factor (TFE3) and the observed inhibition of autophagy.

As above, MEFs lacking TFEB/3 incubated with ChA did not exhibit autophagy defects, nor was there an impact on the activation status of the mTOR effectors evaluated (Figure 3B–H).

### 3.3. Cholesteryl Hemiazelate Causes Apoptosis in Mouse Embryonic Fibroblasts

Due to the critical role of lysosomes and autophagy in cell homeostasis, we next addressed the impact of their failure in MEFs’ quiescence/senescence and cell death. We decided to examine these parameters because lysosome dysfunction induced by ChA caused quiescence/senescence in VSMCs [22] and cell death in macrophages [23]. Cellular senescence can be established by the activation of two different tumor suppressor pathways: CDKN2A (p16)-RB (retinoblastoma) and TP53 (p53)–cyclin-dependent kinase inhibitor 1A (CDKN1A, p21). Thus, we assessed whether these two senescence pathways were activated in MEFs treated with ChA by quantifying the p21 and p16 protein levels by WB. As observed in Figure 4A–C, ChA did not cause an increase in the levels of p21 and p16, indicating that the ChA-treated MEFs were not quiescent/senescent. Then, we decided to evaluate apoptosis by measuring caspase 3 cleavage. Etoposide, an inducer of apoptosis via a caspase-dependent pathway, was used as a positive control. As shown in Figure 4D,E, ChA treatment increased caspase 3 cleavage in a nearly statistically significant manner, indicating that some degree of apoptosis was occurring. Finally, we quantified the lactate dehydrogenase (LDH) activity in ChA-treated MEFs. LDH release is a readout for the loss of the plasma membrane integrity that occurs when the cells undergo secondary necrosis or primary necrosis. As indicated in Figure 4F, the activity of this enzyme in the extracellular culture medium was similar in all experimental conditions, indicating that despite the occurrence of apoptotic cell death, this event was not followed by secondary necrosis. In contrast with the data obtained in ChA-treated WT MEFs, in TFEB/3 DKO cells, no apoptosis was observed.

The occurrence of apoptosis in ChA-treated MEFs can have a strong impact on the pathogenesis of atherosclerosis, since their defective clearance can contribute to inflammation and further cell death [27].

### 3.4. The Presence of TFEB/3 Is Required for the Internalization of Cholesteryl Hemiazelate

As shown in Figure 1G–K, Figure 3B–H and Figure 4A–F, the simultaneous absence of the lysosomal transcription factors TFEB and TFE3 in MEFs exposed to ChA did not have a significant impact on the parameters analyzed (lysosome dysfunction, autophagy, mTOR activation, quiescence/cell senescence and apoptosis). We know from the literature that TFEB/3 DKO cells are resistant to cell death [29]. However, to ensure that the lack of TFEB/3 was indeed responsible for the lack of the phenotypes described above, we decided to address whether the presence of these lysosomal transcription factors was necessary for the uptake of ChA, since the lack of internalization of this lipid could also explain the data obtained above for DKO cells. For this purpose, we started by quantifying the total cholesterol, free cholesterol and cholesteryl esters in WT and DKO MEFs after incubation with ChA for 96 h. Again, POPC was used as a control. As shown in Figure 5A–C, in WT MEFs treated with ChA, the amount of the lipids analyzed per mg of cell protein increased in a significant manner relative to the control cells. These results were expected since ChA, once internalized, can be cleaved inside the lysosomes to free cholesterol and azelaic acid [23]. Additionally, neutral lipid accumulation was observed in ChA-treated MEFs (Figure 1A). In contrast, in DKO MEFs, ChA failed to induce any increase in the levels of the analyzed lipids (Figure 5A–C). These results were a clear indication that the presence of TFEB/3 was required for ChA internalization.

We have previously shown that cholesteryl hemiesters are internalized by passive diffusion and by endocytosis, via the scavenger receptors [46]. Thus, we decided initially to assess whether receptor-mediated endocytosis was impacted in TFEB/3 DKO in comparison to control MEFs (Figure 6A). Based on the uptake of bovine serum albumin (BSA) conjugated with a fluorescent probe (Texas Red), a receptor-mediated process [47], followed for 180 min by flow cytometry in live cultured cells, MEFs were not endocytically very active. MEFs lacking TFEB/3 were even slightly less active than the WT cells (Figure 6A). Since pinocytosis can be an alternative pathway for ChA internalization, we then assessed the impact of the absence of TFEB/3 in the uptake of a fluorescent dextran, a pinocytic probe. As shown in Figure 6B, the internalization of the fluorescent dextran was significantly lower in DKO cells either treated with POPC or ChA in comparison with WT MEFs. Thus, we conclude that TFEB/3 DKO MEFs are less pinocytic than WT MEFs and this outcome could explain, at least in part, the lack of ChA internalization.

Then, we decided to assess the protein levels of two scavenger receptors, SR-BI and CD204 (SRA), which are involved in the uptake of cholesteryl hemiesters in macrophages, in WT and DKO MEFs treated with ChA or POPC for 96 h (Figure 6C–G). The protein levels of the scavenger receptor CD36, which is not involved in the uptake of cholesteryl hemiesters in macrophages [46], as well as the levels of the Toll-like receptor 4 (TLR4), which, upon activation by polyoxygenated cholesteryl ester hydroperoxides, induces robust cytoskeletal rearrangements and macropinocytosis leading to lipid uptake [48,49], were also assessed. For the scavenger receptors SR-BI and CD204 (SRA), a significant increase in their protein levels was observed for WT cells treated with ChA in comparison with POPC-treated WT cells. On the contrary, ChA caused a reduction in CD36 protein levels in WT MEFs. The levels of these scavenger receptors remained similar in ChA- and POPC-treated TFEB/3 DKO MEFs (Figure 6C,F), probably as a result of the lack of ChA internalization. Interestingly, as observed in Figure 6C,D,F in control TFEB/3 DKO MEFs, the protein levels for SR-BI and CD36 were always higher than in control WT MEFs. In contrast, the levels of CD204 (SRA) in TFEB/3 DKO MEFs were lower than in WT MEFs (Figure 6E).

The protein levels of TLR4 were higher in ChA-treated MEFs than in control cells, although the magnitude of the increase was not statistically significant (Figure 6G). For this receptor, the protein levels were similar in WT- and DKO-MEFs treated with POPC or with ChA.

Thus, although the scavenger receptors SR-BI and CD36 are upregulated and CD204 (SRA) is downregulated in TFEB/3 DKO MEFs, the reason(s) that ChA is not internalized warrants further investigation.

## 4. Discussion

Growing interest in fibroblasts, extracellular matrix-producing cells abundant in both the healthy and atherosclerotic vasculature, has been observed [4]. After their activation, some fibroblasts differentiate into myofibroblasts. These activated fibroblasts contribute to extensive remodeling, intimal hyperplasia and luminal stenosis due to their invasion and migration into the intima and the increased production and secretion of ECM proteins [50]. Furthermore, activated fibroblasts increasingly interact, within the arterial wall, with endothelial cells and VSMCs and regulate their functions, as well as recruiting immune cells into the vessel wall [11,51,52].

In the arterial wall, fibroblasts will be exposed to a myriad of different molecules, including oxidized lipids, which can impact their homeostasis, with consequences for the pathology. One of these harmful molecules is ChA, the most prevalent product of LDL-derived cholesteryl ester oxidation [20,21]. This lipid is generated in the arterial intima, and, due to its amphiphilic properties, it is, with time, detected in plasma. The effects of ChA on MEFs, the fibroblasts model used in this study, may be summarized as follows: (1) the formation of lipid-loaded cells with lipid accumulation in the lysosomal compartment; (2) autophagy inhibition; (3) apoptotic cell death.

The effect of ChA at the lysosomal level with consequences for their functioning and accumulation of endocytic cargo, including neutral lipids, was expected. Indeed, based on our previous work, we knew that ChA was sufficient to induce lysosome dysfunction in both macrophages and VSMCs, which, as a consequence, acquired a foamy phenotype [22,23]. The effect of ChA at the lysosomal level is most likely related to the fact that this lipid is an amphiphilic compound, which, due to its cholesterol moiety, partitions very favorably into membranes, increasing their membrane order, similar to the effect of cholesterol [53], causing changes in the v-ATPase activity, an increased luminal pH and decreased acidic hydrolase activity [54]. On the other hand, under physiological conditions, ChA acquires a negative charge, which will confer a different identity and lysosomal membrane composition [23]. Furthermore, ChA inactivates the activity of lysosomal CTSB and cargo egress from this organelle [22].

Curiously, in MEFs, the effect of ChA on lysosomes does not seem to imply the activation of the MiT/TFE TFs, key players in cellular adaptation to lysosome dysfunction. As a consequence, no lysosome mass increase or de novo lysosome biogenesis was observed. This outcome, together with the excessive lipid accumulation in lysosomes, may cause the progressive dysregulation of lysosomes and autophagy function, which can aggravate the deterioration of the plaque microenvironment, accelerating plaque rupture [18]. One possible explanation for the slight activation of only one of the MiT/TFE TFs, TFE3, and autophagy inhibition may be related to the mTOR pathway’s activation, as the increase in 4E-BP, a downstream mTORC1 effector, in the phosphorylated form indicates. Furthermore, in autophagy, macromolecular substances and organelles in the cytoplasm are degraded in autolysosomes, and, to achieve this goal, the presence of functional lysosomes is crucial, and this is not the case in ChA-treated MEFs. Thus, lysosome dysfunction can also contribute to autophagy inhibition, which plays an essential role in maintaining cell homeostasis [55]. Accordingly, defective autophagy accelerates atherosclerosis development by enhancing foam cell formation, inflammation and cell death [39,56,57]. Therefore, it is plausible that autophagy defects, caused by ChA, are responsible for the observed apoptosis of MEFs. The inefficient clearance of apoptotic cells that occurs in atherosclerotic lesions results in secondary necrosis and subsequent inflammation, leading to fibrous cap thinning and necrotic core formation [58,59,60]. Therefore, apoptotic fibroblasts’ death, caused by ChA, can negatively contribute to atherosclerosis.

Increasing evidence indicates a role of TFEB in lipid homeostasis via the inhibition of lipid uptake and an increase in lipid degradation and efflux through mediating lipophagy, lipolysis and lipid metabolism-related genes [31,61,62]. Furthermore, TFEB has been found to also work cooperatively with TFE3 in controlling metabolism, including lipid catabolism, energy metabolism, glucose homeostasis and mitochondrial β-oxidation [63]. Therefore, the absence of these lysosomal transcription factors should, in principle, aggravate the defects in lysosome function and autophagy and increase apoptosis. Unexpectedly, in TFEB/3 double-deficient MEFs, ChA treatment did not cause autophagy or apoptosis. These outcomes are probably related to the fact that ChA is not internalized in TFEB/3 DKO MEFs. We have shown that the internalization of ChE in macrophages occurs by endocytosis after binding to the scavenger receptors SR-BI and CD204 and by partition via passive diffusion and trans-membrane translocation between the ChA-containing particles and the cells [46]. On the other hand, ChA can also be internalized in bulk by micropinocytosis. In TFEB/3 DKO MEFs, the SR-BI and CD36 protein levels are higher than in WT MEFs in basal conditions (results not shown) or when treated with POPC liposomes (ChA vehicle). These results are well correlated with the observations of Yang and collaborators [64], showing that TFEB knockdown in THP-1-derived macrophage exposed to oxidized LDL increased the expression levels of the scavenger receptors. Although the levels of scavenger receptors SR-BI and CD36 are upregulated in TFEB/3 DKO MEFs, this does not lead to the uptake of ChA. For the binding and uptake of ChA, the scavenger receptors must be localized at the plasma membrane. Thus, one of the possible explanations for the failure of ChA internalization is the mis-localization of the scavenger receptors. However, all the commercial antibodies to test this hypothesis, in our hands, were not adequate for immunofluorescence analysis. Alternatively, it is also possible that the main receptor involved in the uptake of ChA in MEFs is the scavenger receptor CD204, and their levels are downregulated in TFEB/3 DKO MEFs. Finally, another possible explanation for the failure of ChA uptake in TFEB/3 DKO cells is the existence of biophysical changes at the level of the plasma membrane that somehow impact the partition via passive diffusion and trans-membrane translocation between the ChA-containing particles and the cells. These biophysical changes can also impact membrane ruffling, which is critical for micropinocytosis to occur. Since micropinocytosis is inhibited in TFEB/3 DKO MEFs, these changes at the level of the plasma membrane are very likely to happen.

## 5. Conclusions

In conclusion, our investigation shows, for the first time, that fibroblasts exposed to ChA, an end-product of the cholesteryl linoleate oxidation found in the tissues of cardiovascular disease patients, exhibit lysosome dysfunction that negatively impacts atherosclerosis due to autophagy inhibition and increased apoptosis. The molecular mechanisms involved in these outcomes need to be further investigated.

## Figures and Tables

**Figure 1 cells-12-02826-f001:**
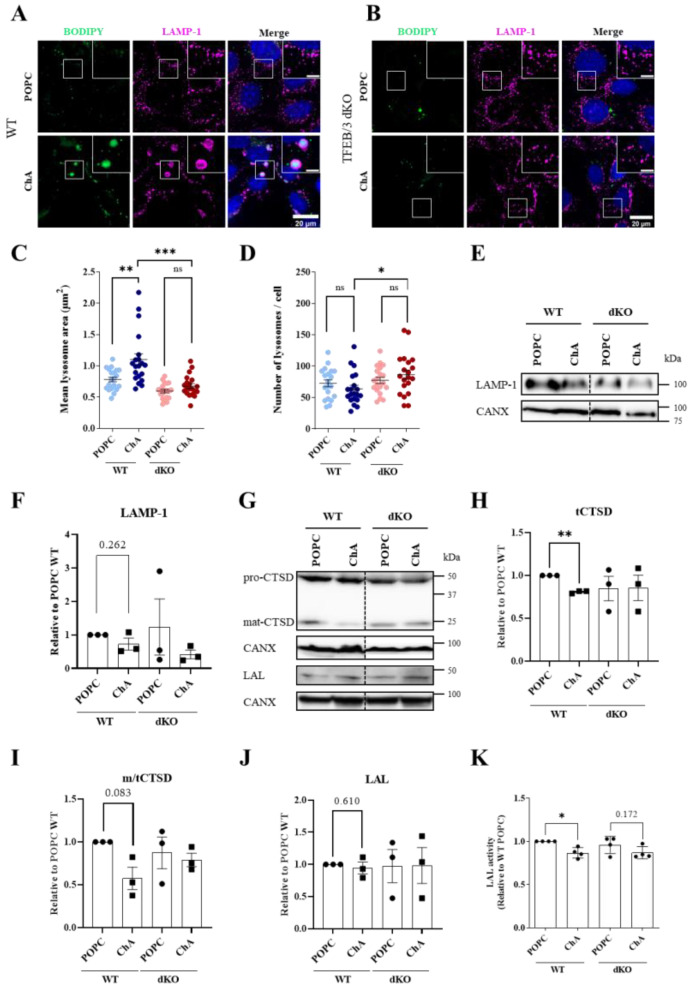
ChA induces lysosomal enlargement in MEFs, in a TFEB/3-dependent manner. (**A**,**B**) MEFs WT (**A**) and TFEB/3-DKO (**B**) were incubated with 269.2 μm of POPC and 500 μm of ChA and, after 72 h, cells were fixed, permeabilized and stained with antibodies against LAMP-1 (magenta, as a lysosomal membrane marker), labelled with Bodipy 493/503 (green, as lipid droplet marker) and DAPI (blue, as nuclear marker). Cells were examined on an Axioimager Z2 microscope (Carl Zeiss) with a 63x NA-1.4 oil immersion objective. Scale bar: 20 μm in the main images and 5 μm in the insets. (**C**,**D**) Quantification of mean lysosome area (**C**) and number of lysosomes per cell (**D**) after 72 h incubation with lipids in WT and TFEB/3-DKO cells. Approximately 20 cells per experimental condition were analyzed and results are presented as the mean ± SEM of three independent biological experiments, ** *p* < 0.01, *** *p* < 0.001, ns, not significant. (**E**) Representative immunoblot of LAMP-1 protein level (calnexin—CANX were used as loading controls) in lysates of MEFs WT and TFEB/3-DKO treated with POPC and ChA for 96 h. (**F**) Quantification of LAMP-1 protein level, normalized to CANX relative to WT cells treated with POPC. Data are presented as mean ± SEM, using a paired *t*-test from three independent experiments. (**G**) Representative immunoblots of cathepsin D (CTSD, pro and mature forms) and lysosomal acid lipase (LAL) protein levels (CANX were used as loading controls) in lysates of MEFs WT and TFEB/3-DKO treated with POPC and ChA for 96 h. (**H**–**J**) Quantification of protein levels, normalized to CANX relative to WT cells treated with POPC. Data are presented as mean ± SEM, ** *p* < 0.01 using a paired *t*-test from three independent experiments. (**K**) Measurement of LAL activity in the lysates of ChA-treated WT and TFEB/3-DKO cells normalized to WT POPC-treated cells. Data are presented as mean ± SEM, * *p* < 0.05 using a paired *t*-test from four independent experiments.

**Figure 2 cells-12-02826-f002:**
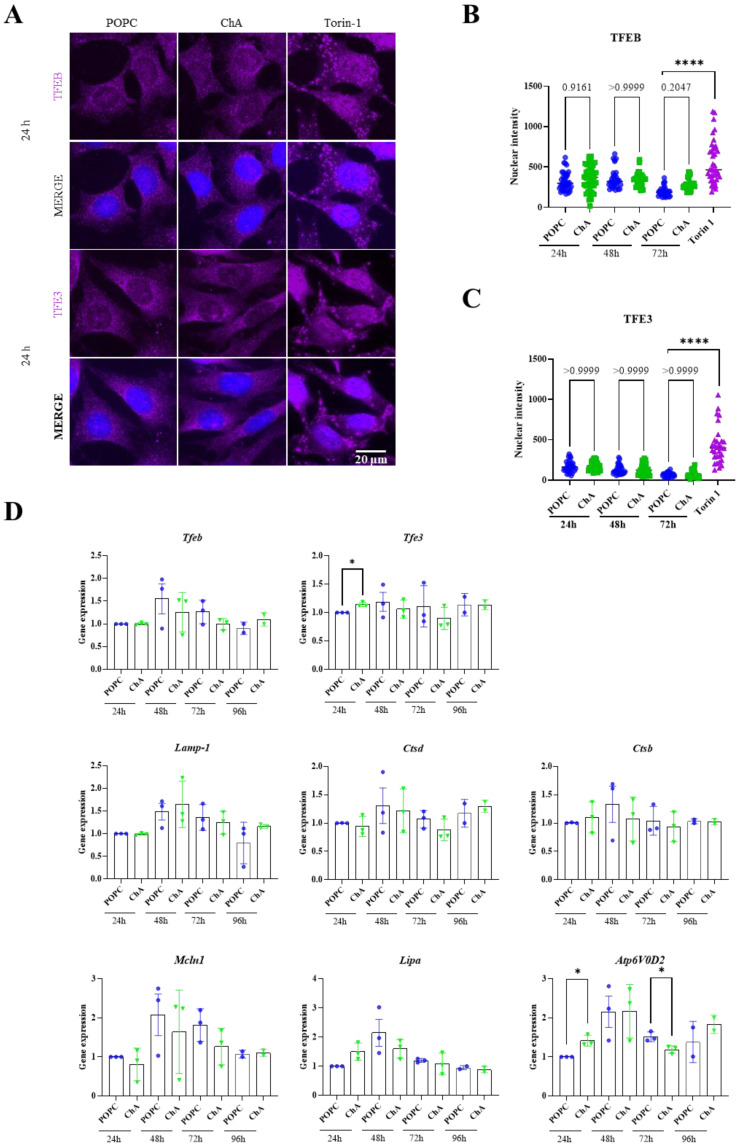
ChA does not induce nuclear translocation of the TFEB and TFE3 transcription factors in MEFs WT cells. (**A**) Representative immunofluorescence images of MEFs WT stained with antibodies against TFEB or TFE3 (magenta), after treatment with 269.2 μM of POPC and 500 μM of ChA for 24 h. Scale bar: 20 μm. (**B**,**C**) Nuclear intensity of TFEF/3 fluorescence at 24 h, 48 h and 72 h and starvation treatment with Torin-1 (positive control) for 4 h, mean ± SEM of three independent experiments, one-way ANOVA test, **** *p* < 0.0001; *n* ≥ 30 cells per condition. (**D**) MEFs WT were treated with lipids for 24 h, 48 h, 72 h and 96 h and qRT-PCR was performed to test mRNA expression of the transcription factors involved in lysosomal biogenesis and autophagy, *Tfeb*, *Tfe3* and the genes controlled by them, *Lamp1*, *Ctsd*, *Ctsb*, *Mcln1*, *Lipa*, *Atp6v0d2*. Values are means ± SEM using a paired *t*-test at the 24 h time point and an unpaired *t*-test for the remaining time points. * *p* < 0.05.

**Figure 3 cells-12-02826-f003:**
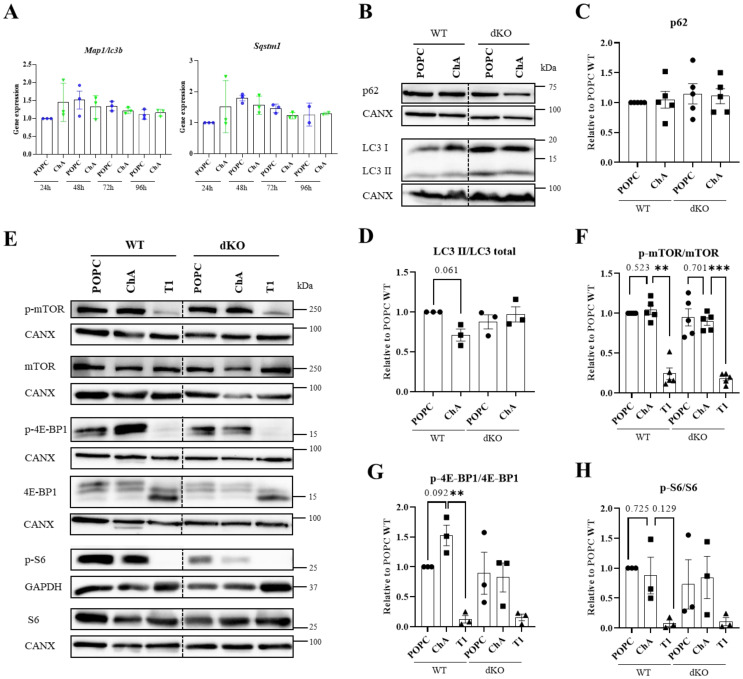
Impact of TFEB/3 on autophagy in MEFs exposed to ChA. (**A**) MEFs WT were treated with 269.2 μM of POPC and 500 μM ChA for 24 h, 48h, 72 h and 96 h and qRT-PCR was performed to test mRNA expression of autophagic genes. (**B**) Representative immunoblots of p62 and LC3 protein levels (CANX were used as loading controls) in lysates of MEFs WT and TFEB/3-DKO treated with POPC and ChA and for 96 h. (**C**,**D**) Quantification of protein levels, normalized to CANX relative to WT cells treated with POPC (*n* = 3 independent experiments). (**E**) Representative immunoblots of phospho-mTOR, mTOR, phospho-4E-BP1, 4E-BP1, phospho-S6 ribosomal protein and S6 ribosomal protein levels (CANX were used as loading controls) in lysates of MEFs WT and TFEB/3-DKO treated with POPC and ChA and for 96 h. (**F**–**H**) Quantification of protein levels, normalized to CANX relative to WT cells treated with POPC. Data are presented as mean ± SEM, ** *p* < 0.01, *** *p* < 0.001, using a paired *t*-test from at least three independent experiments.

**Figure 4 cells-12-02826-f004:**
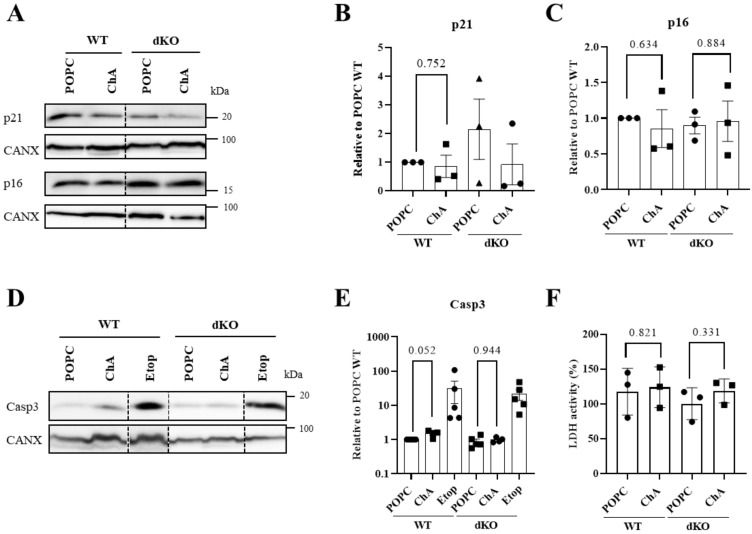
Impact of TFEB/3 on cell proliferation, senescence and cell death. (**A**) Representative immunoblots of p21 and p16 levels (CANX were used as loading controls) in MEFs WT and TFEB/3 DKO lysates after 96 h of treatment with 269.2 μM of POPC and 500 μM of ChA. (**B**,**C**) Quantification of protein levels (shown in (**A**)), normalized to CANX relative to WT cells treated with POPC. Data are presented as mean ± SEM, using paired *t*-test from at least three independent experiments. (**D**) Representative immunoblots of caspase 3 (CANX were used as loading controls) in MEFs WT and TFEB/3 DKO lysates after 96 h of treatment with lipids. (**E**) Quantification of protein levels (shown in (**D**)), normalized to CANX relative to WT cells treated with POPC. Data are presented as mean ± SEM using paired *t*-test from at least three independent experiments. (**F**) Lactate dehydrogenase (LDH) cytotoxicity assay in WT and TFEB/3 DKO MEFs exposed to ChA for 96 h. Data are presented as mean ± SEM, from at least three independent experiments.

**Figure 5 cells-12-02826-f005:**
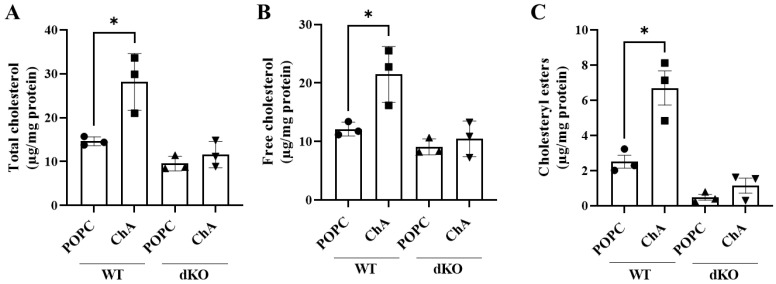
Impact of TFEB/3 on cellular cholesterol metabolism of MEFs exposed to ChA. (**A**–**C**) Total (**A**) and free (**B**) cholesterol levels and cholesteryl esters (**C**) (microgram per milligram of cell protein) were determined by fluorometric quantification in MEFs WT and TFEB/3 DKO lysates after 96 h of treatment with 269.2 μM of POPC and 500 μM of ChA. Data are from three independent experiments presented as mean ± SEM, * *p* < 0.05, using unpaired *t*-test.

**Figure 6 cells-12-02826-f006:**
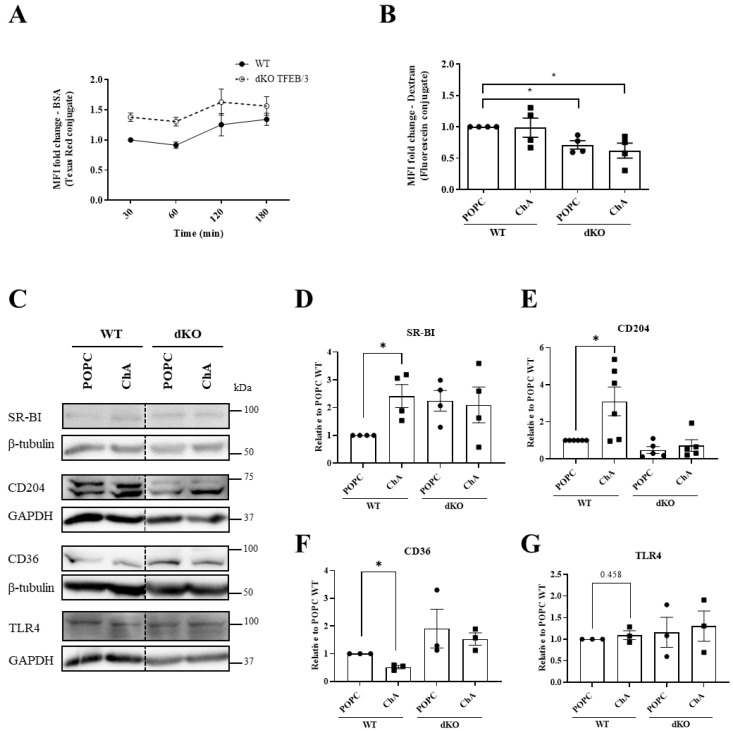
Impact of TFEB/3 on receptor-mediated endocytosis and micropinocytosis. (**A**,**B**) WT and TFEB/3 DKO MEFs were incubated with BSA–Texas Red for different time points (**A**) or with dextran–fluorescein for 24 h in presence of POPC or ChA (**B**). The internalization of both fluorescent probes was measured by flow cytometry. Results are presented as the mean fluorescence intensity (MFI) ± SEM of four independent biological experiments, * *p* < 0.05, using a paired *t*-test. (**C**) Representative immunoblots of scavenger receptors SR-BI, CD204 and CD36 and TLR4 (GAPDH and β-tubulin were used as loading controls), in MEFs WT and TFEB/3 DKO lysates after 96 h of treatment with 269.2 μM of POPC and 500 μM of ChA. (**D**–**G**) Quantification of protein levels (shown in **C**), normalized to GAPDH or β-tubulin relative to WT cells treated with POPC. Data are presented as mean ± SEM, * *p* < 0.05, using a paired *t*-test from at least three independent experiments.

**Table 1 cells-12-02826-t001:** Primer sequences.

Gene	Sequence	
	Forward	Reverse
*Tfeb*	AGGAGCGGCAGAAGAAAGAC	CAGGTCCTTCTGCATCCTCC
*Tfe3*	CCTGAAGGCATCTGTGGATT	TGTAGGTCCAGAAGGGCATC
*Lamp-1*	ACATCAGCCCAAATGACACA	GGCTAGAGCTGGCATTCATC
*Ctsd*	GCTTCCGGTCTTTGACAACCT	CACCAAGCATTAGTTCTCCTCC
*Ctsb*	TCCTTGATCCTTCTTTCTTGCC	ACAGTGCCACACAGCTTCTTC
*Mcln1*	GCGCCTATGACACCATCAA	TATCCTGGCACTGCTCGAT
*Lipa*	CTAGAATCTGCCAGCAAGCC	AGTATTCACCGAATCCCTCG
*Atp6v0d2*	CAGAGCTGTACTTCAATGTGGAC	AGGTCTCACACTGCACTAGGT
*Map1/lc3*	GACGGCTTCCTGTACATGGTTT	TGGAGTCTTACACAGCCATTGC
*Sqstm1*	5GTCTTCTGTGCCTGTGCTGGAA	TCTGCTCCACCAGAAGATCCCA
*Pgk1*	ATGGATGAGGTGGTGAAAGC	CAGTGCTCACATGGCTGACT
*Gapdh*	GGGAAGCCCATCACCATCTTC	AGAGGGGCCATCCACAGTCT

## Data Availability

The data that support the findings of this study are available from the corresponding authors, A.R.A.M. and O.V.V., upon reasonable request.

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
