# Peer review of "Cholesteryl Hemiazelate Present in Cardiovascular Disease Patients Causes Lysosome Dysfunction in Murine Fibroblasts"

_cells, 2023, doi:10.3390/cells12242826_

Round 1

Reviewer 1 Report

Comments and Suggestions for Authors

Overall the study is clearly presented and of general interest. I have some concern about the quality of the western blots.

(Fig. 1E, G) Fig. 3B, Fig. 4A and D, and Figure 6C). Some data of Blots are shown and quantified, the controls (CANX) are often overexposed and bands are not clearly visible. Some bands (e.g. TLR4 Fig 6C, Fig 1E LAMP) show no clear bands, sometimes with high background. Since there are sometimes significant differences, I would suggest to re quantify the blots and repeat some blots. Why was sometimes CANX used and sometimes GAPDH or tubulin?

I would like to see in the discussion the differences of fibroblasts from human compared to the used mouse fibroblasts MEF.

Author Response

Overall the study is clearly presented and of general interest. I have some concern about the quality of the western blots.

We would like to thank the reviewer for reading the manuscript and providing constructive feedback.

(Fig. 1E, G) Fig. 3B, Fig. 4A and D, and Figure 6C). Some data of Blots are shown and quantified, the controls (CANX) are often overexposed and bands are not clearly visible.

We can assure the reviewer that none of the western blots presented in the manuscript are overexposed. The blots were all developed using the “optimal exposure” function of the ChemiDoc system. The blots have had minor contrast enhancement as provided by the ImageLab (Biorad(R)) software (image for publication option), perhaps that is what the reviewer is referring to. When using the ChemiDoc/ImageLab system, overexpossed blots appear as red pixels on the captured chemiluminescence picture. We can nonetheless present the images without contrast enhancement if that remains an issue. All the loading control images as far as we can see show clearly visible bands.

Some bands (e.g. TLR4 Fig 6C, Fig 1E LAMP) show no clear bands, sometimes with high background.

TLR4 and LAMP1 are both heavily glycosylated proteins and, as such, on western blot they do not present as one very defined band but rather as a “smear” representing the different molecular weights caused by the different glycosylation patterns. As the reviewer is probably aware, not all antibodies have the same quality and not all protein are expressed at high levels in every cell type. Our TLR4 western blots may indeed be considered to have a relatively high background due to the quality of the antibody in question and the relative low expression of the protein in MEFs. A quick search for antibodies against murine TLR4 shows that our blots resemble those provided by the manufacturer's images. In addition, in our hands, the antibody against murine TLR4 provides much clearer blots when using macrophage samples because TLR4 is much more abundant in these cells.

Since there are sometimes significant differences, I would suggest to re quantify the blots and repeat some blots. Why was sometimes CANX used and sometimes GAPDH or tubulin?

Different proteins are used as loading controls depending on the percentage of the SDS-PAGE gels used for the analysis. High molecular weight proteins, such as CANX are less well transferred in high percentage gels (e.g., 15%), so it is preferable to use lower MW proteins such as GAPDH and tubulin. When using a low percentage acrylamide gel (e.g., 10%), high molecular weight loading controls like CANX are the standard choice.

I would like to see in the discussion the differences of fibroblasts from human compared to the used mouse fibroblasts MEF.

We are not aware of any study comparing MEFs with human fibroblasts. However, large-scale transcriptomic analyses, at single-cell resolution, comparing mouse and human fibroblasts “side-by-side” suggest the existence of analogous fibroblast subtypes and similar gene activation programs between these two species (doi: 10.1038/s41586-021-03549-5; doi: 10.1038/s41467-022-30633-9, doi: 10.1681/ASN.2019040335; doi: 10.1038/s41467-021-22331-9). Thus, based on the literature, we do believe that the studies performed in MEFs will be of relevance for human translation.

Reviewer 2 Report

Comments and Suggestions for Authors

The authors Lopes et. al, have submitted an original manuscript titled "Cholesteryl hemiazelate present in cardiovascular disease patients causes lysosome dysfunction in murine fibroblasts". The article discusses the effects of cholesteryl hemiazelate treatment on fibroblasts and elucidates the molecular mechanism of this interaction. The study is almost complete one requiring some minor revisions -

Since the authors do not use patient-derived ChA they should reword their title as "Cholesteryl hemiazelate causes lysosome dysfunction via a TFEB/3 axis in murine fibroblasts."

On line 108 and at other places in the manuscript, the authors have referred to their control cells as WT Null MEFs. The term "Null" in these locations is misleading and the authors should refer to these cells as just "WT MEFs" unless they have specific characteristics that require these cells to be called "WT Null MEFs".

The authors need to include a catalog number of Lalistat2.

The description of results from lines 252 - 256 is repetitive and can be combined into a single sentence.

On line 256, the authors should revise their description of the assay to immunofluorescence instead of fluorescence as they have used antibodies against LAMP1.

On lines 258 - 259, the authors should support their interpretation of the cause of neutral lipid accumulation in lysosomes with similar results from their other studies or other literature showing such data.

On lines 256 - 257, the authors refer to the BODIPY stained structures in the lysosomes as neutral lipid accumulation while they refer to neutral lipid structures in WT cells as lipid droplets. The authors should justify their differential use of nomenclature. Also, lipid droplets exist in different sizes. Therefore, the authors need to justify why the structures within the lysosomes could be something different from lipid droplets.

On line 265, the authors need to cite an article apart from their studies that have used LAMP1 expression levels as an indicator of "lysosome mass". 

In the results of TFEB/3 dKO MEF treatment with ChA, the authors have incorrectly placed the statistically significant indicators in Figure 1C. Also, the authors do observe a significant increase in the numbers of lysosomes in ChA-treated dKO cells vs WT (Figure 1D). However, the authors have not described this in the text. Also, it is surprising that though the authors see an increase in the number of lysosomes, they do not see an increase in LAMP1 protein levels in these respective groups.

In the description for Figure 1I, the authors claim that 50% of cathepsin was in its inactive form in ChA-treated WT cells. However, in the figure, the authors have quantified the ratio of mature cathepsin to total cathepsin. Therefore, a more accurate description of the results would be that the ChA-treated WT cells had a 50% lower efficiency of processing as compared to untreated WT cells. To comment on the levels of the inactive form of the enzyme, the authors would need to calculate a ratio of pro-cathepsin to total cathepsin.

On lines 296 - 297, the authors need to cite figures 1G-K for the results of the dKO cells.

On lines 337 - 340, a more accurate justification for measuring the expression levels of Tfeb and Tfe3 mRNA would be - "To confirm that ChA was affecting nuclear translocation and not expression of these TFs, we decided to quantify the mRNA levels of Tfeb/3".

The authors do not need to justify the increase in Tfe3 mRNA levels at 24h as there might be an increase in intensity of the staining of TFE3 in their immunofluorescence studies. Also, as pointed out by them, despite an increase in expression levels, there is no nuclear translocation of TFE3. Therefore, the expression level increase has no effect in this context.

The statement on line 370 about the effect of autophagy inhibition on atherosclerosis is out of context and better be discussed further and included in the discussion section.

On line 371, the authors mention that the absence of TFEB/3 doesn't significantly impact autophagy. While this observation is true, the result can be presented in the following way to enhance the impact of the results - "The inhibition of autophagy upon ChA treatment was lost in Tfeb/3 dKO cells indicating that these TFs play a role in autophagy induction as well."

The results of LDH activity erroneously cited Figure 4C instead of Figure 4F.

The representative image for dKO cells in Figure 4D does not represent the statistics in Figure 4E. The authors could replace the representative image.

In lines 513 to 515, the authors mention that the reason for the lack of ChA internalization in dKO cells with increased SR-BI and CD204 levels warrants further investigation. However, their results in Figures 6A and 6B show that MEFs use pinocytosis and not scavenger receptors (like SR-BI, CD204) mediated endocytosis for their uptake. Therefore, increased expression of these scavenger receptors may not have any effect on the levels of ChA uptake.

The effect of ChA appears to be limited to a population of lysosomes and not all of them in the cell. This population is the one that is specifically increasing in size due to ChA treatment (Figure 1B and outliers in Figure 1C). Surprisingly they can affect the cathepsin maturation across all lysosomes. The authors could include this point in the discussion.

Author Response

The authors Lopes et. al, have submitted an original manuscript titled "Cholesteryl hemiazelate present in cardiovascular disease patients causes lysosome dysfunction in murine fibroblasts". The article discusses the effects of cholesteryl hemiazelate treatment on fibroblasts and elucidates the molecular mechanism of this interaction. The study is almost complete one requiring some minor revisions -

We would like to thank the reviewer for reading the manuscript with such great care and for all the constructive feedback provided. We feel that the suggestions given substantially helped us to improve the quality of the manuscript.

Since the authors do not use patient-derived ChA they should reword their title as “Cholesteryl hemiazelate causes lysosome dysfunction via a TFEB/3 axis in murine fibroblasts.”

The ChA used in this study is chemically similar to the one that we have previously described in CVD patient plasma and atherosclerotic plaques, therefore we feel that the present title is appropriate. We also feel that our results concerning the role of the TFEB/3 transcription factors in the described phenotype require future investigation before we feel confident enough to make such a statement in the title.

On line 108 and at other places in the manuscript, the authors have referred to their control cells as WT Null MEFs. The term "Null" in these locations is misleading and the authors should refer to these cells as just "WT MEFs" unless they have specific characteristics that require these cells to be called "WT Null MEFs".

We agree with the reviewer and have removed the term “Null” from the text and figures.

The authors need to include a catalog number of Lalistat2.

We have added the catalogue number of Lalistat2 to line 198.

The description of results from lines 252 - 256 is repetitive and can be combined into a single sentence.

We agree with the reviewer and have therefore combined the sentences to read as follows:

“As shown in Fig. 1A (see insets) and C, ChA causes a statistically significant enlargement of the area of lysosomes near the perinuclear region when compared with the lysosome area in POPC-treated cells.”

On line 256, the authors should revise their description of the assay to immunofluorescence instead of fluorescence as they have used antibodies against LAMP1.

We agree and have replaced the term “fluorescence” by “microscopy” in this sentence.

On lines 258 - 259, the authors should support their interpretation of the cause of neutral lipid accumulation in lysosomes with similar results from their other studies or other literature showing such data.

Accordingly, we have added references 30 and 31 to fundament the statement on lines 258-259.

On lines 256 - 257, the authors refer to the BODIPY stained structures in the lysosomes as neutral lipid accumulation while they refer to neutral lipid structures in WT cells as lipid droplets. The authors should justify their differential use of nomenclature. Also, lipid droplets exist in different sizes. Therefore, the authors need to justify why the structures within the lysosomes could be something different from lipid droplets.

BODIPY is routinely used to mark lipid droplets. We distinguished lysosomes from lipid droplets based on the immunostaining for the lysosomal membrane protein LAMP-1. To clarify this criterium we have added the following information to the sentence: “In contrast, in POPC exposed cells, there were no detectable BODIPY-positive lysosomes (LAMP1-positive structures) and just a limited number of small lipid droplets (LAMP1-negative structures, Fig. 1A, upper panels), indicating that neutral lipids are not being stored under these conditions.”

On line 265, the authors need to cite an article apart from their studies that have used LAMP1 expression levels as an indicator of "lysosome mass".

We have added the reference Tol et al. Autophagy 2018 to line 267.

In the results of TFEB/3 dKO MEF treatment with ChA, the authors have incorrectly placed the statistically significant indicators in Figure 1C.

The reviewer is correct, thank you for noticing this. We have corrected the Fig. 1C.

Also, the authors do observe a significant increase in the numbers of lysosomes in ChA-treated dKO cells vs WT (Figure 1D). However, the authors have not described this in the text. Also, it is surprising that though the authors see an increase in the number of lysosomes, they do not see an increase in LAMP1 protein levels in these respective groups.

We added the following sentence to the paragraph to address the legitimate point raised by the reviewer (line 277): “However, we did detect a minor (barely significant) increase in lysosome number in ChA-treated DKO cells compared to ChA-treated WT cells (Fig. 1D), not reflected by the total LAMP1 protein level (Fig. 1E).” Please keep in mind that this is an extremely marginal and barely significant increase in lysosome number, which perhaps explains why it is not reflected at the total LAMP1 protein level assessed by western blot.

In the description for Figure 1I, the authors claim that 50% of cathepsin was in its inactive form in ChA-treated WT cells. However, in the figure, the authors have quantified the ratio of mature cathepsin to total cathepsin. Therefore, a more accurate description of the results would be that the ChA-treated WT cells had a 50% lower efficiency of processing as compared to untreated WT cells. To comment on the levels of the inactive form of the enzyme, the authors would need to calculate a ratio of pro-cathepsin to total cathepsin.

The reviewer is once again correct in this observation, we apologize for the lapse. We have accordingly corrected the statement to: “As observed in Fig. 1G and quantified in Fig. 1H, the total cathepsin D levels were lower in ChA-treated cells than in control cells and roughly 50 % less of this lysosomal enzyme was efficiently processed to the active (mature) form (Fig. 1I).”

On lines 296 - 297, the authors need to cite figures 1G-K for the results of the dKO cells.

We have added the reference to the Fig.1G-K to this sentence (now line 300).

On lines 337 - 340, a more accurate justification for measuring the expression levels of Tfeb and Tfe3 mRNA would be - "To confirm that ChA was affecting nuclear translocation and not expression of these TFs, we decided to quantify the mRNA levels of Tfeb/3".

We followed the suggestion of the reviewer and changed the sentence on line 341 accordingly.

The authors do not need to justify the increase in Tfe3 mRNA levels at 24h as there might be an increase in intensity of the staining of TFE3 in their immunofluorescence studies. Also, as pointed out by them, despite an increase in expression levels, there is no nuclear translocation of TFE3. Therefore, the expression level increase has no effect in this context.

We agree with the reviewer and have removed this justification from line 353-355 accordingly.

The statement on line 370 about the effect of autophagy inhibition on atherosclerosis is out of context and better be discussed further and included in the discussion section.

We have removed this sentence since a similar statement is made in the Discussion section.

On line 371, the authors mention that the absence of TFEB/3 doesn't significantly impact autophagy. While this observation is true, the result can be presented in the following way to enhance the impact of the results - "The inhibition of autophagy upon ChA treatment was lost in Tfeb/3 dKO cells indicating that these TFs play a role in autophagy induction as well."

We gladly accepted the suggestion of textual alteration made by the reviewer.

The results of LDH activity erroneously cited Figure 4C instead of Figure 4F.

Thank you for alerting us to this lapse.

The representative image for dKO cells in Figure 4D does not represent the statistics in Figure 4E. The authors could replace the representative image.

We have replaced the representative blots shown for the dKO cells in Fig. 4D, nonetheless we would like to alert the reviewer to the fact that the graph shown in Fig. 4E has a logarithmic scale in the y-axis which may influence the visual perception of these small differences.

In lines 513 to 515, the authors mention that the reason for the lack of ChA internalization in dKO cells with increased SR-BI and CD204 levels warrants further investigation. However, their results in Figures 6A and 6B show that MEFs use pinocytosis and not scavenger receptors (like SR-BI, CD204) mediated endocytosis for their uptake. Therefore, increased expression of these scavenger receptors may not have any effect on the levels of ChA uptake.

Although MEFs are not endocitically very active the SR are upregulated in WT-MEFs treated with ChA. This outcome is similar to what has been described in the literature for cells treated with oxidized LDL. In those experimental settings, the increase in the SRs expression, has been linked to unregulated uptake of oxidized LDL and to foam cell formation. In our opinion, the results obtained in our experimental settings, are an indication that SR can also be involved in the ChA uptake in MEFs.

The effect of ChA appears to be limited to a population of lysosomes and not all of them in the cell. This population is the one that is specifically increasing in size due to ChA treatment (Figure 1B and outliers in Figure 1C). Surprisingly they can affect the cathepsin maturation across all lysosomes. The authors could include this point in the discussion.

We do not have data showing that ChA impacts cathepsin maturation across all lysosomes. The data on cathepsins was obtained by WB since the antibodies available do not distinguish the mature from the immature form. The dysfunctional enlarged lysosomes are localized mainly in the perinuclear region of MEFs treated with ChA. This phenotype is like that observed in VSMCs treated with this same lipid (Alves et al. JCS 2022) and to that described for lysosomal storage diseases (LSD) (doi: 10.1083/jcb.201208152).
